# The Importance of Fatty Acids as Nutrients during Post-Exercise Recovery

**DOI:** 10.3390/nu12020280

**Published:** 2020-01-21

**Authors:** Anne-Marie Lundsgaard, Andreas M. Fritzen, Bente Kiens

**Affiliations:** Section of Molecular Physiology, Department of Nutrition, Exercise and Sports, Faculty of Science, University of Copenhagen, 2100 Copenhagen, Denmark; amlundsgaard@nexs.ku.dk (A.-M.L.); amfritzen@nexs.ku.dk (A.M.F.)

**Keywords:** post-exercise recovery, fatty acid oxidation, skeletal muscle, lipid metabolism, molecular mechanism, adipose tissue lipolysis, AMP-activated protein kinase (AMPK), pyruvate dehydrogenase (PDH), carnitine palmitoyltransferase I (CPT1), lipoprotein lipase (LPL)

## Abstract

It is well recognized that whole-body fatty acid (FA) oxidation remains increased for several hours following aerobic endurance exercise, even despite carbohydrate intake. However, the mechanisms involved herein have hitherto not been subject to a thorough evaluation. In immediate and early recovery (0–4 h), plasma FA availability is high, which seems mainly to be a result of hormonal factors and increased adipose tissue blood flow. The increased circulating availability of adipose-derived FA, coupled with FA from lipoprotein lipase (LPL)-derived very-low density lipoprotein (VLDL)-triacylglycerol (TG) hydrolysis in skeletal muscle capillaries and hydrolysis of TG within the muscle together act as substrates for the increased mitochondrial FA oxidation post-exercise. Within the skeletal muscle cells, increased reliance on FA oxidation likely results from enhanced FA uptake into the mitochondria through the carnitine palmitoyltransferase (CPT) 1 reaction, and concomitant AMP-activated protein kinase (AMPK)-mediated pyruvate dehydrogenase (PDH) inhibition of glucose oxidation. Together this allows glucose taken up by the skeletal muscles to be directed towards the resynthesis of glycogen. Besides being oxidized, FAs also seem to be crucial signaling molecules for peroxisome proliferator-activated receptor (PPAR) signaling post-exercise, and thus for induction of the exercise-induced FA oxidative gene adaptation program in skeletal muscle following exercise. Collectively, a high FA turnover in recovery seems essential to regain whole-body substrate homeostasis.

## 1. Introduction

Exercise activities, especially of longer duration, cause large quantities of energy to be expended. This necessitates metabolic recovery processes to restore substrate stores in recovery. A plethora of metabolic changes occur in order to regain substrate homeostasis, especially in the previously active skeletal muscles, but also in other organs such as adipose tissue and the liver. These processes are energy demanding and accompanied by enhanced post-exercise oxygen consumption (EPOC), of which the magnitude is dependent on exercise duration and intensity [1]. Besides EPOC, the relative contribution from fatty acid (FA) oxidation for whole-body energy turnover is increased, demonstrated by a decreased respiratory exchange ratio (RER), for several hours following aerobic exercise in both untrained and endurance-trained individuals [2,3,4,5,6,7,8,9], and notably also in obese and type 2 diabetic subjects [10]. Even with a carbohydrate-rich meal intake in recovery, the contribution of FAs to oxidation is still increased when compared with resting control conditions [2,4]. Notably, a prolonged effect of aerobic exercise on substrate metabolism is documented by lower RER values persisting to the following morning after prior exercise compared with prior rest [4,5,11].

The regulatory mechanisms for the increased FA oxidation post-exercise—which likely reflects increased FA oxidation in skeletal muscle—could reside both at the level of circulating lipid availability as well as metabolic regulation within skeletal muscle. The substrates fueling enhanced FA oxidation during recovery from exercise seem to be an interdependent mixture of FAs from circulating albumin-bound FAs and FAs from hydrolysis of both plasma triacylglycerol (TG) and myocellular TG (IMTG). The physiological significance of the maintained elevation in FA oxidation might be to favor the resynthesis of skeletal muscle glycogen stores from the available plasma glucose. In addition, increased FA availability during recovery from exercise might also induce important signaling in skeletal muscle.

In the present review, physiological and molecular mechanisms underpinning the enhanced FA availability and the elevated FA oxidation in skeletal muscle during post-exercise recovery will be addressed, while we also aim to discuss the concomitant gluco- and lipid metabolic consequences of the increased FA availability in recovery.

## 2. Increased Circulating Fatty Acid Availability in Recovery from Exercise

Within the first hour after an aerobic exercise bout of low to moderate-intensity, an immediate increase in plasma FA concentration to 1600–2100 µmol/L has been observed in numerous studies [6,7,9,12,13,14] (Figure 1A). The magnitude of this increase in plasma FA concentration in immediate recovery, from the level achieved during the prior exercise bout, is positively related to exercise intensity [13].

The immediate peak in plasma FA concentration in the hour after termination of exercise is transient, but beyond this, several studies have shown that circulating FA levels remain elevated at ~600–1000 µmol/L for 2–6 h into recovery when compared with resting pre-exercise levels [6,12,13,15,16]. Significantly elevated plasma FA concentrations have been observed even at 20–42 h post-exercise [7,11,17]. Notably, increased fasting plasma FA values were also obtained in the morning following exercise compared with prior rest, when carbohydrate-rich meals were consumed on the prior exercise day [4,7,11]. Accordingly, when exercise in non-trained men and women was performed in the afternoon, followed by ingestion of an evening meal comprising 55 energy% (E%) carbohydrate, fasting plasma FA rate of appearance, measured by a labeled palmitate infusion the next day, was found to be elevated both at 13–16 h (40%) and 21–24 h (10%) post-exercise, when compared with resting control conditions [17]. The finding of an increased plasma glycerol concentration for as much as 24 h after prolonged exercise at 70%–75% of maximal oxygen consumption rate (VO_2_peak) [5] further supports that whole-body lipolysis remains elevated in late recovery.

Collectively, the increased FA availability following aerobic exercise can be divided into 3 periods; (1) immediate recovery, a peak in the plasma FA concentration at ~1600–2100 µmol/L in the first 60 min, (2) early recovery, increased plasma FA concentration of ~600–1000 µmol/L in the first 1–4 h, (3) late recovery, sustained elevated fasting or postabsorptive plasma FA concentration at 10%–40% above similar conditions with prior rest, despite intake of carbohydrate-rich meals or not. An intriguing question is which regulatory mechanisms are underlying the immediate (0–1 h), early (1–4 h), and longer-term (>4 h) elevation in plasma FA concentration post-exercise.

### 2.1. Immediate Recovery

The peak in circulating FA concentration in the immediate recovery period following submaximal exercise has been suggested to be due to a delayed spill-over effect from the increased adipose tissue lipolysis induced during exercise. As the exercise-induced increase in adipose tissue blood flow has been shown to be less than the exercise-induced increase in adipose tissue lipolysis rate [18], a fraction of the lipolytic liberated FAs may be entrapped in adipocytes during exercise and thus released at exercise stop. However, adipose tissue blood flow has been shown to return to pre-exercise levels immediately at termination of exercise [19,20], which makes a post-exercise washout of entrapped FAs less likely Alternatively, the transient peak in plasma FA levels immediately after exercise may rather be explained by a more rapid reduction in plasma FA uptake in skeletal muscle [21] due to discontinuation of muscle contractions.

### 2.2. Early Recovery

The whole-body lipolytic rate, determined from the plasma glycerol rate of appearance, increases by up to 400% above resting values during exercise of 1 to 4 h at 40%–65% of VO_2_peak [6,12,21], likely reflecting an increased lipolytic rate in adipose tissue. Following exercise, plasma glycerol rate of appearance decreases immediately following exercise [6,12,21] (Figure 1B), while remaining elevated above resting conditions in early fasted recovery, shown by a greater adipose tissue glycerol and FA release at 2–3 h post-exercise compared with rest, when directly measured in a subcutaneous abdominal vein [19,22]. Thus, enhanced adipose tissue lipolysis in early recovery likely plays an important role in the sustained elevated circulating FA levels during the first hours following exercise.

Mechanisms responsible for the increased adipose tissue lipolysis compared with rest in the hours after exercise stop may be related to hormonal regulation [23]. Accordingly, when well-trained men performed cycling exercise at 70% of the Watt-max test until exhaustion (~20 min) with the β-adrenergic blocker, propranolol, plasma levels of FA and glycerol were markedly reduced at exercise stop and in the first 20 min of recovery [24]. This suggests that catecholamine-induced beta-adrenoceptor mediated adipose tissue lipolysis during and in close proximity after exercise is important for the remaining increase in circulating levels of FAs. While norepinephrine and epinephrine have a short half-life in plasma (few minutes), they are first observed to have fully returned to pre-exercise levels within ~15–120 min post-exercise, depending on the intensity of the prior exercise bout and training status of the subjects [13,15,16,25,26]. Circulating catecholamines may, therefore, contribute to the high FA levels in the early recovery (up to 2 h), but beyond that, catecholamines are not likely to regulate the increased adipose tissue lipolysis.

Growth hormone (GH) concentrations are increased in plasma during prolonged aerobic exercise in humans in a linear fashion with exercise intensity [27] and remain elevated in the first hour into recovery [15,22]. When GH was infused into humans under resting conditions, it appeared that the stimulating effect on adipose tissue lipolysis was delayed 1–2 h [28,29,30,31,32]. GH secreted during exercise may, therefore, have delayed effects on adipose tissue lipolysis into recovery. Supporting this notion, it was observed in trained young and moderately trained older men that GH concentrations remained increased 1–2 h into recovery from 20 min of exercise at 70% of VO_2_peak [15]. Furthermore, when 1 h of either GH infusion or exercise were performed, similar high circulating FA concentrations and glycerol rate of appearance were obtained during a subsequent 4 h period [15], suggesting that the GH induction during exercise increases adipose tissue lipolysis during early recovery. In support, when GH secretion was abolished by administration of the somatostatin analog octreotide, from 1 h before, during 1 h of exercise at 50% of VO_2_peak, and 4 h post-exercise, plasma FA concentration, and glycerol Ra rates were blunted during the 4 h post-exercise period [22]. Thus, the exercise-induced increase in plasma GH concentration appears to play a role in increased adipose tissue lipolysis in early recovery. However, there seems to be a gender-specific regulation, since plasma GH was not found to be significantly increased by 1 h of exercise at 65% VO_2_peak or 30–180 min into recovery from this exercise bout in female in contrast to male subjects [16].

Circulating insulin levels potently inhibit adipose tissue lipolysis [33]. Plasma insulin concentrations are decreased during exercise and remain lower in early recovery compared with pre-exercise or resting conditions until glucose or meal ingestion [4,7,9,34]. Thus, a lower plasma insulin concentration can also contribute to an increased adipose tissue lipolytic rate in early recovery.

Besides hormonal regulation, adipose tissue blood flow might play a role in the release of FA during early recovery. Adipose tissue blood flow, measured by the ^133^Xe washout technique, was increased ~50%–180% during whole-body exercise at 40%–70% of VO_2_peak [19,20,35]. After the immediate reduction of subcutaneous abdominal blood flow after the termination of exercise, a progressive increase has been shown during 3 h post-exercise recovery to levels obtained during exercise [19]. Thus, increased adipose tissue blood flow might contribute to increased adipose tissue FA release during the first hours of recovery. It remains unassessed whether adipose tissue blood flow remained elevated compared to resting conditions later in recovery.

In summary, the regulatory mechanisms mediating increased adipose tissue lipolysis and circulating FA levels in early recovery (up to 4 h) likely involve a combination of: (1) Delayed effects of β-adrenoceptor-mediated lipolysis initiated during exercise, (2) increased plasma GH and lower plasma insulin concentrations during recovery, and (3) increased adipose tissue blood flow during recovery. The elevated plasma FA concentration in the early hours after aerobic exercise increased FA availability for uptake and oxidation within the previously exercised skeletal muscles, while also increasing FA delivery to other tissues such as the liver.

The mechanisms explaining the increased plasma FA and glycerol concentrations observed in the late recovery phase [4,7,11,17] are not well established. Studying the isolated effect of prior exercise on lipid metabolism in the late recovery is difficult since factors as exercise-induced energy deficit and meal intake during recovery will interfere. However, when the exercise-induced energy deficit was compensated for by an intake of meals following exercise on the exercise day, elevated overnight-fasted FA levels, and lower RER values were still evident the day following exercise [4]. Furthermore, afternoon exercise led to increased nocturnal FA oxidation despite the intake of evening meals [36]. Thus, exercise seems to affect substrate utilization independently of exercise-induced energy deficit.

## 3. Plasma VLDL-TG as an Additional FA Source during Exercise Recovery

In addition to FA availability from adipose tissue lipolysis, hydrolysis of very low-density lipoprotein (VLDL)-TG might contribute with FAs to oxidation in skeletal muscle post-exercise.

Lower fasting plasma TG levels have been reported in both early (2 h) [37] and late (12 h) [38] recovery from exercise lasting 1.5–2 h at 50%–60% of VO_2_peak in moderately trained men and women. Even at 42 h to 66 h following prolonged high-intensity exercise, fasting plasma TG levels of trained and untrained individuals were reduced by 27%–39% compared with resting conditions [39,40].

Lipoprotein lipase (LPL), located at the luminal site of the endothelial cells in the capillary bed of peripheral tissues, is responsible for the hydrolysis of core TG in VLDL and chylomicrons prior to FA uptake by target tissues [41]. A delayed increase in the activity of LPL in skeletal muscle has been found 3–6 h after prolonged exercise in trained men during fasted recovery, independent of insulin-stimulation [42,43] or intake of carbohydrate-rich meals [7]. One study also reported increased muscle LPL activity in late recovery (30 h), when carbohydrate-rich meals were consumed [7]. Muscle LPL protein content has been found to be increased at 8 to 22 h after exercise stop [44,45,46], why an increased availability of muscle LPL protein could contribute to the increased muscle LPL activity later in recovery.

Collectively, the presence of enhanced LPL activity in skeletal muscle capillaries in the post-exercise period suggests an enhanced hydrolytic capacity of circulating TG-rich lipoproteins, further enhancing the availability of FAs for skeletal muscle. In support of this notion, the plasma VLDL-TG concentration was decreased by ~40% during 4.5 h of recovery following 90 min of moderate-intensity exercise in moderately trained men, when compared with a non-exercise trial [47]. Moreover, a higher fractional catabolic rate of plasma VLDL-TG, measured by tracer techniques, has been reported after exercise and to gradually increase up to 3–4 h into recovery in young, sedentary female and male subjects after 45 min of moderate-intensity exercise (40% VO_2_peak) [48]. Even at 12 h after 2 h exercise at 60% of VO_2_peak followed by an evening meal, overnight-fasted plasma VLDL-TG clearance rate was found to be increased compared with prior rest in untrained men [38]. Increased clearance of plasma lipid fractions was also observed at postprandial conditions later post-exercise, as plasma clearance of total TG, chylomicron-TG, and VLDL-TG across the leg was increased following a mixed meal provided 14 h after a 2 h running bout at 64% of VO_2_peak [49]. Such reductions in postprandial lipemia in the hours after an aerobic exercise bout have consistently been observed and found to be dependent on the volume and intensity of the prior exercise bout [50].

Taken together, it appears that that circulating TG from lipoproteins, in addition to albumin-bound FAs, contribute as important nutrients for skeletal muscle FA oxidation during recovery and that mechanisms regulating both adipose tissue lipolysis and VLDL-TG hydrolysis secure the availability of circulating FAs in recovery from prior exercise. Basal muscle blood flow is slightly increased in the first hours of recovery while returning to resting levels at 4 h post-exercise [51,52]. Sustained post-exercise vasodilation in the first hours following exercise, found primarily in the previously exercised musculature, is proposed to be mediated by combined central neural mechanisms (altered arterial baroreflex settings reducing sympathetic activity despite a lower arterial pressure) and local vasodilatory mechanisms [53]. In addition, when exercise was performed with one leg and the other served as non-exercised control, postprandial skeletal muscle blood flow, measured in the femoral vein, remained higher in the prior exercised leg when measured up to 8 h post-exercise (Kiens et al., unpublished). Furthermore, postprandial femoral blood flow was higher following prior exercise compared to rest when measured 17 h post-exercise [49]. Increased postprandial blood flow within skeletal muscle into recovery from exercise will further contribute to increased FA availability for uptake upon nutrient intake.

## 4. Potential Mechanisms Regulating Fatty Acid Oxidation in Skeletal Muscle

### 4.1. Fatty Acid Uptake into the Skeletal Muscle

Almost 100% of the fraction of plasma FAs that are taken up in the leg in the first 3 h into recovery from one-legged exercise are oxidized, and only a negligible fraction of labeled tracer FAs are found to be re-esterified to IMTG during fasting post-exercise conditions [21]. Coinciding with the low RER values in early recovery, oxidation of plasma-derived FAs is increased in the first 3 h and represents the major part of the FAs oxidized during the early recovery period [21]. This suggests that membrane transport of plasma FAs into the skeletal muscle is increased in recovery. Such a phenomenon could be regulated simply by the increased availability of plasma FAs but may also include myocellular transport mechanisms. Thus, despite the fact that FAs can easily enter and diffuse across cellular membranes, trans-sarcolemmal FA transport involves membrane-associated FA binding proteins, in which cluster of differentiation 36/SR-B2 (CD36) is a key facilitator [54]. During exercise, CD36 in skeletal muscle translocates to the sarcolemma. This is shown by subfractionation of mouse muscle during treadmill running [55], and in giant sarcolemmal vesicles (GSV) obtained from human muscle, in which a 75% increase in CD36 protein content after prolonged submaximal exercise was observed [56]. Other FA transporters are also translocated during exercise to the plasma membrane, such as plasma membrane fatty acid binding protein (FABPpm) and FA transport protein (FATP) 1 and FATP4 [56,57]. Manipulation of their protein content in rodent skeletal muscle modified FA uptake under resting conditions [58,59]. One could speculate that the contraction-induced relocation of FA transporters to the membrane persists into recovery from exercise and thereby facilitates a high transmembrane FA uptake in this period. However, the role of these FA transport proteins and their myocellular localization in recovery from exercise are not understood and should be a topic for future investigations. CD36 is known to translocate to sarcolemma upon insulin-stimulation in rat skeletal muscle [60], but whether an increased sensitivity to insulin e.g., in relation to a carbohydrate-rich meal is apparent for CD36 translocation in human skeletal muscle in recovery remains to be determined.

A recent study described another potential player in the regulation of FA uptake and oxidation in skeletal muscle in relation to exercise. Thus, the acute administration of the circulating lipid 12,13-dihydroxy-9Z-octadecenoic acid (12,13-diHOME) in mice in vivo increased skeletal muscle FA uptake and oxidation [61]. Interestingly, a bout of moderate-intensity exercise increased 12,13-diHOME in both sedentary and trained men and women, an increase which remained for several hours into recovery [61]. This exercise-induced 12,13-diHOME lipokine could contribute to the increased FA oxidation in skeletal muscle in early recovery, and the regulatory mechanisms remain to be investigated further in humans.

Importantly, however, arteriovenous FA sampling and stable isotope-labeled palmitate measurements across the leg in recovery from 2 h one-legged knee extensor exercise at 65% of maximal leg power output [21] suggested that uptake of albumin-bound FAs into human skeletal muscle declines within the first hours of recovery (though still remaining higher than at resting conditions), while leg FA oxidation remained at 44%, increased compared with resting levels. This could argue that FAs derived from hydrolysis of plasma VLDL-TG or TGs located in cytoplasmic lipid droplets in skeletal muscle (IMTG) constitute additional FA sources during post-exercise recovery.

### 4.2. The Role of IMTG Metabolism in Skeletal Muscle FA Oxidation in Recovery

One study directly measured post-exercise IMTG synthesis by labeled palmitate infusion over 2 h following 90 min of moderate-intensity exercise, and found IMTG fractional synthesis rates to be negative in endurance-trained subjects in the fasted state [10]. This suggests that IMTG is not synthesized in early fasted recovery, but rather can contribute as an energy substrate.

When IMTG content was measured biochemically in muscle biopsies, one study, performed in endurance-trained men, showed IMTG content to be unchanged compared with the post-exercise value measured over 18 h of recovery from 90 min exhaustive endurance exercise [9]. However, in that study, RER was also only lower than pre-exercise the first hour following exercise [9]. Another study in trained men showed IMTG content to be lower compared to both pre- and post-exercise values at both 3, 6, and 18 h in recovery from exhaustive endurance exercise [7]. Importantly, the extent to which IMTG is utilized during exercise and recovery seems to depend on the muscle fiber type, since reductions in IMTG content were only obtained in type I and not in type IIa muscle fibers during [62] and 4 h [63] into recovery from a 2–4 h bicycle exercise bout at 56%–75% of VO_2_peak.

In all, this could suggest hydrolysis of IMTG and the contribution of the liberated FAs to oxidation during recovery. In the studies detecting hydrolysis of IMTG during recovery, carbohydrate-rich, low-fat meals were ingested post-exercise [7,63], suggesting that insulin-induced suppression of plasma FA availability plays a role in the relative utilization of plasma FAs versus IMTG-derived FAs. To this end, when post-exercise IMTG content was assessed in trained men 24 h after 2 h of moderate-intensity exercise, followed by either high-carbohydrate, low-fat (83E%, 5E%) or low-carbohydrate, high-fat (16E%, 68E%) meal intake on the exercise day, IMTG content was reduced compared with both pre- and post-exercise values only in the high-carbohydrate fed group [64]. Thus, the extent to which IMTG contributes to increased muscle FA oxidation during recovery seems to depend on plasma FA availability.

In all, post-exercise data suggest that hydrolysis of IMTG in skeletal muscle is evident in an interdependent manner with plasma FA availability, and potentially also the degree of additional hydrolysis of LPL-derived plasma VLDL- or chylomicron-TG. Together, these different FA substrates enhance the availability of FAs for the skeletal muscle cells in recovery. Whether the increased availability of FAs is the main drive for increased FA oxidation or regulation within the skeletal muscle is primary in mediating the increased FA oxidation, which can next be questioned.

### 4.3. Myocellular FA Handling

Cytosolic FAs, originating from either plasma or intramyocellular lipolysis, must be activated to fatty acyl-CoAs by a family of acyl-CoA synthetases (ACSs) located at the plasma membrane, mitochondria, and lipid droplets for further metabolism. The isoform ACS1 seems crucial for partitioning FAs towards oxidation in skeletal muscle during exercise, as mice with muscle-specific ACSL1 deletion had lower FA utilization during treadmill endurance exercise than control mice [65]. The specific role of ACS in the regulation of FA oxidation during recovery has not been clarified.

For fatty acyl-CoAs to be oxidized in the mitochondria, the carnitine palmitoyltransferase (CPT) system is important. The CPT system transports FA into mitochondria, a rate-limiting step for acetyl-CoA production by FA in the β-oxidation. Following FA activation by ACS in the cytosol, fatty acyl-CoAs are carnitinized to enter the mitochondria for oxidation, a reaction that requires carnitine and the enzyme CPT1, located at the outer mitochondrial membrane. Since carnitine is a substrate in the CPT1 reaction, changes in the free carnitine content in skeletal muscle during recovery may contribute to the regulation of FA import and hence FA oxidation. Besides being a substrate for CPT1, free carnitine also acts as an acceptor of acetyl groups, thereby forming acetyl-carnitine and free CoA. This carnitine entrapment is expected to be increased when acetyl-CoA is generated in excess of its metabolism in the TCA cycle, which appears especially at high glycolytic rates [54]. Interestingly, skeletal muscle acetyl-carnitine content was, together with acetyl-CoA, decreased markedly during the first 3–6 h of recovery from 90 min exhaustive exercise in endurance athletes [9]. The large reduction in skeletal muscle acetyl-carnitine content early in recovery is probably due to a reduced glycolysis rate, resulting in more free carnitine available for cytosolic FAs, thereby increasing the supply of fatty acyl-CoA for β-oxidation. The reduced glycolysis rate could arise from PDH inhibition or increased direction of glucose-6-phosphate (G6P) to glycogen resynthesis due to increased glycogen synthase activity—or more likely a combination of these.

Malonyl-CoA, the product of the acetyl-CoA carboxylase (ACC) reaction, is a regulator of CPT1 activity [66,67] and thereby contributes to the control of FA entrance and oxidation in the mitochondria. In rat skeletal muscle, ACC activity and malonyl-CoA content remained suppressed, and FA oxidation enhanced 90 min after 30 min treadmill exercise, indicating that the ACC-malonyl-CoA-CPT1 axis post-exercise adds to the increased FA entrance into mitochondria and thus FA oxidation [68]. In humans, we have shown a ~30% reduction in malonyl-CoA content relative to resting levels in skeletal muscle at 4 h fasted recovery from 1 h one-legged knee extensor exercise at 80% of peak workload, concomitantly with a sustained increase in inhibitory ACC phosphorylation [69]. This further supports a decreased inhibition of CPT1 in recovery, allowing for a greater influx of fatty acyl-CoA to the mitochondria for oxidation.

Of note, post-exercise FA oxidation is enhanced following unaccustomed eccentric exercise [70,71]. This phenomenon is associated with findings of lower muscle glucose uptake and glycogen resynthesis, persistent to several days following the exercise bout [72]. This implies that substrate oxidation in recovery may depend on exercise mode, via altered glucose utilization.

## 5. Muscle Glucose Oxidation and Glycogen Resynthesis during Recovery—A Role of AMPK

The preference for FA oxidation in the post-exercise period in the skeletal muscle implies a downregulation of glucose oxidation. Pyruvate dehydrogenase (PDH) appears to be important in such a mechanism. PDH catalyzes the conversion of pyruvate from glycolysis to acetyl-CoA through oxidative decarboxylation, why inhibition of PDH activity will lead to decreased glucose oxidation. The control of PDH activity is complex. Accordingly, the products of the PDH reaction (acetyl-CoA, NADH) exert feedback inhibition on its activity. PDH activity is also covalently controlled by pyruvate dehydrogenase kinases (PDK). PDK4 is one of four isoenzymes that phosphorylates and inactivates PDH, thereby inhibiting the conversion of pyruvate to acetyl-CoA [61]. PDK4 mRNA expression in skeletal muscle is increased when circulating FA availability is increased, as, e.g., during a high-fat diet [73]. Interestingly, PDK4 mRNA has been shown to be rapidly upregulated by 10-fold in human skeletal muscle after 60–90 min exhaustive one-legged knee-extensor exercise and remains upregulated for 4 h into recovery [74]. Furthermore, a reduction of ~50% in PDH activity was obtained in human skeletal muscle over 6–18 h after 90 min exhaustive exercise in endurance-trained athletes [9]. Together, these findings point to a reduction in skeletal muscle PDH activity in recovery from exercise, indicating a decreased conversion of pyruvate into acetyl-CoA and thereby a reduction in the contribution of glycolysis-derived acetyl-CoA in the TCA cycle.

Studies in mice lacking the catalytic subunit of the energy sensor, AMP-activated protein kinase(AMPK)α_2_ in skeletal muscle have shown higher RER values (i.e., lower FA oxidation) during the first 6 h of recovery following 2 h of treadmill exercise compared with wild-type (WT) mice [75]. This was observed concomitantly with a 65% upregulation of PDK4 protein at the cessation of exercise in WT, but not in AMPKα_2_ knock-out (KO) mice, which was obtained together with a lower inhibitory PDH phosphorylation and lower NADH content (the latter mediating allosteric inhibition of PDH and activation of PDK) [75]. AMPK regulation of PDK4 in skeletal muscle is supported by increased transcription of PDK4 after treatment with the AMPK activator, the AMP-analogue 5-Aminoimidazole-4-carboxamide ribonucleotide (AICAR), in primary rat cardiomyocytes [76]. Together, these findings point to that activation of AMPK during exercise is important for PDK4 induction and the inactivation of PDH leading to suppression of glucose relative to FA oxidation in skeletal muscle in recovery.

The oxidation of FA to cover energy expenditure in skeletal muscle during recovery from exercise is likely a metabolic priority of skeletal muscle to synthesize muscle glycogen stores from the available glucose [7,77,78]. The direction of FA within skeletal muscle towards oxidation during post-exercise seems to be governed by mechanisms within skeletal muscle.

AMPK might play a dual regulatory role at this point by increasing FA oxidation, but also by promoting glycogen synthesis directly. To this end, we have observed 22% lower skeletal muscle glycogen content 6 h post-exercise in AMPKα_2_ KO compared with WT mice, despite similar glycogen content at the termination of exercise [75]. Other findings in mice lacking either functional AMPKα_2_ or AMPKγ_3_ also showed lower skeletal muscle glycogen levels compared with WT in recovery after 30–90 min of treadmill running [79,80] and 2 h of swimming exercise [81]. Together these data implicate that AMPK is important in partitioning glucose for muscle glycogen resynthesis rather than oxidation during recovery. Even more directly, it was recently shown in a muscle-specific, inducible AMPK catalytic isoform KO mouse model (AMPKα1/α2 imdKO) that muscle glycogen synthesis was markedly reduced when glucose was administered orally following exercise [82]. Furthermore, AMPK was recently shown to be important for muscle basal glucose uptake [83] and insulin-stimulated glucose uptake [84] in mice following muscle contractions and in vivo exercise via regulation of TBC1D1 [83] and TBC1D4 [85], respectively. Collectively, this suggests that muscle glycogen synthesis is of high metabolic priority during recovery, and that AMPK might be an important player in facilitating this substrate navigation indirectly via the regulation of substrate oxidation and/or directly via regulation of glycogen synthesis and/or glucose uptake.

## 6. Molecular Metabolic Adaptations in Skeletal Muscle during Recovery

The increased FA availability and/or increased FA oxidation during recovery from exercise might induce signaling in skeletal muscle, in turn affecting the metabolic response to exercise. Accordingly, when high-fat and low-carbohydrate meals were ingested during recovery from moderate-intensity exercise in trained male subjects, the mRNA contents of several exercise-responsive metabolic genes, such as PDK4, uncoupling protein 3 (UCP3), LPL, CPT1, CD36, forkhead homolog in rhabdomyosarcoma 1 (FOXO1), and peroxisome proliferator-activated receptor gamma coactivator 1α (PGC-1α), were increased 8–24 h into recovery [86]. In contrast, when a carbohydrate-rich diet was consumed during recovery, reducing plasma FA levels to ~15% of the values in the high-fat trial, no such delayed elevation appeared in the mRNA content of these genes [86]. Likewise, intake of ~60 g carbohydrate within 3 h of recovery attenuated the exercise-induced increase in PDK4, UCP3, and GLUT4 mRNA levels observed in fasted recovery in moderately trained subjects [87], concomitantly with 260% lower FA levels compared with fasted recovery. This point to a more potent effect on exercise-induced gene transcription during low carbohydrate/high fat intake or fasting in recovery, likely related to the greater circulating FA availability. FAs, and in particular unsaturated FAs, provide ligands and hence activate peroxisome proliferator-activated receptor (PPAR)α and PPARδ, thereby transcribing genes involved in lipid metabolism. It has been shown that FAs isolated from human plasma are ligands for the PPARs [88]. Accordingly, when nicotinic acid was administrated prior to exercise, leading to marked suppression of plasma FA levels, nicotinic acid was found to blunt the exercise-induced increase in PGC-1α, PPARα, and PPARδ mRNA contents observed in the exercised control group [89], proving the role of intramuscular FA availability for the exercise-induced gene response. More directly, it has been shown that the treatment of mice with a PPARδ-agonist resulted in a dose-dependent activation of FA oxidation in skeletal muscle, supported by higher gene expression of β-oxidation enzymes, FA transport proteins, UCP3, and LPL [90]. In addition, mice lacking PPARδ in skeletal muscle demonstrated lower exercise FA oxidation following 4 weeks of exercise training compared with WT mice [91], further suggesting an important role of PPARδ in the training-induced lipid metabolic protein machinery. Since the PPARδ promoter is found to be hypomethylated at 3 h into recovery from acute exercise compared with resting conditions in humans, concomitant with increased PPARδ mRNA expression [92], FA-induced DNA modification could be an early event in post-exercise-induced lipid metabolic gene activation.

These findings suggest that FA-induced PPAR signaling is pivotal in the regulation of the FA oxidation program in skeletal muscle during recovery from exercise. Hereby, nutrients seem to regulate their own metabolism.

To further support an effect of FA oxidation for the adaptive response to exercise training, overexpression of a malonyl-CoA insensitive CPT-1 mutant in mouse muscle increased muscle FA oxidation and increased expression of lipid metabolic genes such as PPARβ, CD36, PDK4 [93].

## 7. Concluding Remarks

Whole-body FA oxidation is increased for several hours following aerobic exercise, even with carbohydrate-rich meal intake during recovery from exercise. Plasma FAs liberated from the adipose tissue, LPL-derived VLDL-TG hydrolysis in the capillaries surrounding the skeletal muscle, and hydrolysis of IMTG within the muscle together act as substrates in an interdependent manner and enhance the availability of FA for the skeletal muscle cells. The high FA oxidation in skeletal muscle during recovery from exercise appears to be regulated at several steps in the skeletal muscle cell, including an enhanced FA uptake into the mitochondria through the CPT1 reaction. Moreover, a simultaneously AMPK-mediated PDH inhibition of glucose oxidation allows high FA oxidation to occur. Collectively, this allows the glucose taken up to be directed towards the resynthesis of muscle glycogen stores. The elevated level of FA within the myocytes could be crucial signaling molecules for e.g., PPAR signaling in the regulation of the exercise-induced FA oxidation program in skeletal muscle during recovery from exercise.

## Figures and Tables

**Figure 1 nutrients-12-00280-f001:**
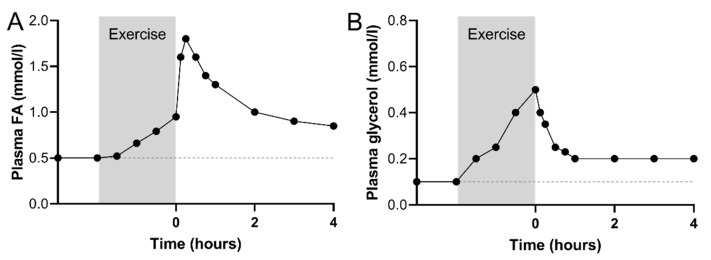
Representative illustration of plasma fatty acid (FA) (**A**) and glycerol (**B**) concentrations during moderate-intensity exercise (80 min to 4 h at 40%–70% of maximal oxygen consumption rate (VO_2_peak)) and in early recovery in endurance-trained subjects. Subjects remained fasted in recovery. The scatter plots are based on data in references [3,6,12,13,18].

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
