# Peer review of "The Importance of Fatty Acids as Nutrients during Post-Exercise Recovery"

_nutrients, 2020, doi:10.3390/nu12020280_

Round 1

Reviewer 1 Report

Nutrients-manuscript reviews: MDPI

The importance of  fatty acids as nutrients during post-exercise recovery

By Lundsgaard et al

This review supplies well information about the metabolism of fatty acids and glucose during the process of recovery of exercise and its regulation involved. It is carefully organized and presented for the related information with various aspects from some animal experiment and clinical trials. Some critical genes such as CPT, PPAR, PGC1 and AMPK were also described. It is well organized and presented and very informative for a broad audience of researchers on metabolism, nutrients and health related.

However, although it is well organized for the general contents, very few references are after 2017 (4 out of 76).  Some very recent progress on the energy substrate homeostasis and hormone regulation related with exercise should be included.

Minor comment:

Page 3, Line 97 …”Ra” should be in full name for clarification.

Page 3, Line 112 …”watt-max…”, should be “…Watt-max test…”  

Reviewer 2 Report

It is an interesting review; however, there were some issues that were not discussed. One key issue is the fiber type metabolism and the fiber type modulation depending on the exercise and endurance training. The increase in vascular tone, lower peripheral resistance, and high metabolic demand needs to be addressed. 

The other issue is How epigenetic modulation may affect fatty acid metabolism? May lipid accumulation occur in damaged muscles?

The article by article Tsiloulis T, Watt MJ. Exercise and the Regulation of Adipose Tissue Metabolism. Prog Mol Biol Transl Sci. 2015;135:175–201, may be important to discuss. 
